# Interleukin-13 (IL-13)—A Pleiotropic Cytokine Involved in Wound Healing and Fibrosis

**DOI:** 10.3390/ijms241612884

**Published:** 2023-08-17

**Authors:** Elke Roeb

**Affiliations:** Department of Gastroenterology, Justus Liebig University Giessen, Klinikstr. 33, 35392 Giessen, Germany; elke.roeb@innere.med.uni-giessen.de; Tel.: +49-641-985-42338

**Keywords:** NAFLD, NASH, IL-13, liver, fibrosis, metabolism, MASLD, MASH

## Abstract

The liver, as a central metabolic organ, is systemically linked to metabolic–inflammatory diseases. In the pathogenesis of the metabolic syndrome, inflammatory and metabolic interactions between the intestine, liver, and adipose tissue lead to the progression of hepatic steatosis to metabolic-dysfunction-associated steatohepatitis (MASH) and consecutive MASH-induced fibrosis. Clinical and animal studies revealed that IL-13 might be protective in the development of MASH through both the preservation of metabolic functions and Th2-polarized inflammation in the liver and the adipose tissue. In contrast, IL-13-associated loss of mucosal gut barrier function and IL-13-associated enhanced hepatic fibrosis may contribute to the progression of MASH. However, there are only a few publications on the effect of IL-13 on metabolic diseases and possible therapies to influence them. In this review article, different aspects of IL-13-associated effects on the liver and metabolic liver diseases, which are partly contradictory, are summarized and discussed on the basis of the recent literature.

## 1. Introduction

This review article uses the new nomenclature for steatotic liver disease (SLD) announced by the AASLD and EASL, including MASLD (formerly NAFLD), MASH (NASH), and MASH fibrosis [1]. In many diseases that are primarily classified as chronic inflammatory (e.g., asthma, arthritis, autoimmune skin diseases, type-2-diabetes, non-alcoholic liver disease), sub-types which are characterized by metabolic comorbidities have been described in recent years. These comorbidities include, in particular, metabolic syndrome, dysregulation of fat and carbohydrate metabolism, and/or associated cardio-vascular diseases. Prominent examples include obese asthmatics and obese patients with fatty liver, inflammatory bowel disease, or patients with allergic contact dermatitis and obesity [2,3,4]. These associations between chronic inflammatory disease processes and metabolic dysregulation have led to the introduction of the term “metaflammation” [5,6]. In this context, “metaflammation” should be understood as the link between metabolism and inflammation [7]. Metabolic–immunological interactions in chronic and acute inflammation and infectious diseases, as well as immune–metabolic and peri-tumoral interactions in malignant diseases, are to be distinguished from this.

The prevalence of metabolic-dysfunction-associated steatotic liver disease (MASLD) has risen sharply [8,9,10]. The highest prevalence is currently found in Latin America, at 44.4%. In Western Europe, 25.1% of the population suffers from MASLD [10]. Overall, the worldwide prevalence of steatotic liver disease is estimated at about 30% [10]. But currently, there is no approved pharmacotherapy for metabolic-dysfunction-associated steatohepatitis (MASH), the most common liver disease in the Western world and developing countries [11,12]. Patients with MASLD can develop metabolic-disease-associated steatohepatitis (MASH) and progression to MASH-fibrosis and -cirrhosis. Liver transplantation is limited by the small number of available donor organs, the risk of surgical intervention, and the risk of immune rejection post-transplantation. Consequently, the development of alternative strategies for anti-fibrotic and anti-inflammatory liver regeneration is urgently needed with a view to clinical application [13,14].

Interleukin-13, a cytokine with various functions in the immune system, has structural and functional similarities with interleukin-4. IL-13 acts a messenger substance that is involved in processes of the immune system, especially in the triggering of allergic reactions [15]. The proinflammatory cytokine IL-13, which was identified in 1989, has a very broad spectrum of action. A recent review [16] explains the importance and function of IL-13 for the regulation of immunoglobulins (Ig), inflammation, anti-parasitic reactions, fibrogenesis, and allergic reactions. IL-13 can also be released as a preform by granulocytes such as basophils, mast cells, or eosinophils, a process in which IgE plays an important role [16,17]. IL-13 binds to the IL-13Rα1 receptor, which then recruits the IL-4Rα to the type 2 receptor complex. Any blockade of IL-13 signal transduction is thus possible either by preventing binding to the IL-13Rα1 receptor or the IL-4Rα receptor [16,18,19]. Figure 1 illustrates the several receptors and signaling pathways of IL-13 according to [20].

IL-13 is suspected of being the main mediator for triggering asthma attacks [16,21]. At the cellular level, IL-13 is a mediator of the humoral immune reaction (antibody production by B cells). In this process, it is produced by TH2 helper cells and stimulates the differentiation of B lymphocytes [22]. IL-13 also inhibits the activation of macrophages and induces matrix metallo-proteinases (MMPs), e.g., in the respiratory tract and gingival fibroblasts [16,23,24]. IL-13 acts as a pleiotropic cytokine. On the one hand, increased IL-13 expression can elicit several proinflammatory effects [16,25,26]. On the other hand, IL-13 functions as a mediator of the humoral immune response, is produced by TH2 helper cells, and stimulates the differentiation of B lymphocytes [27]. Thus, IL-13 stimulates a wide range of innate and adaptive immune cells, as well as non-hematopoietic cells, to coordinate various functions, including immune regulation, antibody production, and fibrosis [16,26]. In a murine sepsis model, IL-13 protected mice from lethality, and an IL-13 blockade decreased survival from peritonitis [25].

In schistosomiasis, IL-13 has emerged as a central mediator of chronic-infection-induced liver fibrosis and portal hypertension, and IL-13 was identified as a primary mediator of liver fibrosis [28,29]. However, in the studies performed on female C57BL/6 mice, TGFβ as well as IL-17 and IL-13 were elevated in the murine serum [29]. The neutralization of IL-17 alone also led to a decrease in IL-13 and TGFβ. It thus remains largely unclear in these animal studies what effect IL-13 alone has on fibrosis induction by schistosomiasis. Intrahepatic innate lymphoid cells are involved in the modulation of homeostatic and inflammatory processes in various tissues. Recently, IL-13-producing ILC3-like cells that were enriched in the human liver were shown to be involved in the modulation of chronic liver disease [26]. For this exciting work, tissue material from liver resections was used as well as explanted organs from patients with viral hepatitis, alcohol toxic liver cirrhosis, MASLD, primary biliary cholangitis, and primary sclerosing cholangitis. The authors did not use organs from patients with schistosomiasis. It thus remains unclear and to be evaluated whether this particular subgroup of IL-13-producing ILC3-like cells also occurs in the context of *S. mansoni*-induced liver fibrosis. Fibroblasts were recently identified to be an IL-13-responsive cell type [30]. But, in this context, IL-13 acts more or less indirectly with the help of IL-33 [31]. In IL-33-receptor knockout mice, e.g., the serum levels of IL-13 and IL-17, both profibrotic cytokines, were significantly lower than in wild-type mice. Also, the α-smooth muscle actin expression was lower in chronic schistosomiasis in mice lacking IL-33 signaling compared to the controls [32]. The question arises whether IL-13 plays a key role in linking inflammation and metabolism especially in metabolic liver diseases, e.g., MASLD. This pleiotropic cytokine is involved in the progression of steatosis to MASH but also has anti-inflammatory effects. Obesity-associated inflammation in adipose tissue, for example, can be normalized by IL-13 [33]. The hydrodynamic injection of the IL-13 gene was able to completely prevent Western-diet-induced obesity in the experiments conducted here. The side effects of obesity, such as insulin resistance or hepatic steatosis, were also prevented [33]. However, the gene therapy approach used here is unsuitable in the translational setting and limits clinical applicability. The mechanistic effects of IL-13 on energy metabolism at the cellular and molecular level need further evaluation. On the other hand, IL-13 was associated with biliary fibrosis, diarrhea, and perianal inflammation in a genetic model of cholestasis, ABCB4-knockout mice [34]. These results seem to be contradictory or at least need clarification. The main question here is whether different fibrosis models (metabolic versus biliary-induced fibrogenesis), the age or sex of the animals, and different mouse strains exert an influence on IL-13 effects.

## 2. IL-13 in Wound Healing Processes

A few decades ago, Ahdieh et al. examined the barrier function in lung epithelial cells. Using wound healing assays, they showed that IL-13 reduced wound healing and the maintenance of barrier function [35]. A disadvantage of this older work is the restriction of the investigations to a human adenocarcinoma cell line from a lung tumor. Only the direct effects of cytokines and interferons on growth behavior in vitro could be investigated [35]. The innate or adapted immune system was completely left out of this early work.

A recent review summarizes the information known so far about IL-13, signaling pathways, and the functional effects of this pleiotropic cytokine [36]. The differences between IL-4 and IL-13, which share transmembrane receptor molecules (illustrated in Figure 1), were elaborated. In a schistosome model of liver fibrosis, IL-13 inhibition led to granuloma formation and increased survival [37,38]. Similarly, in a pulmonary fibrosis model, IL-13 antagonization reduced fibrosis and collagen deposition [39]. There are pronounced functional differences between IL-4 and IL-13 with regard to allergic inflammation, worm infections, and fibrogenesis [36]. Granuloma-building eosinophils seem to be a source of IL-13 [36,40]. By producing profibrotic mediators and polarizing the Th2 response, eosinophils play an important role in schistosomiasis-induced liver fibrosis [36,41]. Inhibiting eosinophils might thus be a player with regard to improving chronic fibrotic diseases [36,42]. Evolutionarily conserved innate lymphoid cells (ILCs), which include not only natural killer (NK) cells and lymphoid-tissue-inducer (LTi) cells but also cells that produce IL-13, act as regulators of immunity and tissue remodeling [43]. In this context, a new family of innate lymphoid cells (ILC) has been identified which has been shown to produce IL-13 associated with T helper type 2 responses [43]. The ILC system is probably an ancient system that predates the adaptive immune system [44].

For several years, it has been known that IL-13 is involved in cardiac wound healing and remodeling after myocardial infarction (MI). Hofmann et al. demonstrated that IL-13 expression contributes not only to monocyte differentiation but improved survival after experimental MI in mice. Interleukin-13 improved myocardial healing and remodeling at least in male mice [45]. In airway epithelial cells, the IL-13 signals via IL-13Rα2 to mediate repair, which depends on the HB-EGF-dependent activation of EGFR. On the other side, dysregulated IL-13 signaling in the airways of asthmatics contributes to the epithelial barrier dysfunction observed in asthma [46]. IL-13 acts on fibrotic skin diseases in a comparable manner. For these entities, IL-13 might reveal a potential target for novel therapies with regard to prevention or treatment [47]. Excessive wound reactions lead to so-called skin fibrosis. The resulting scarring is extremely stressful and also functionally disturbing. The serum levels of IL-13 were upregulated in patients with systemic scleroderma compared with healthy controls. In addition, the immunoregulatory Th2 cytokine IL-13 also mediates important profibrotic effects in the skin. IL-13 induces, e.g., the proliferation of dermal fibroblasts and collagen synthesis in skin cells [47].

Recently, using a statistical approach, it has been shown that IL-13 in combination with matrix stiffness is able to regulate macrophage morphology, M2 polarization profile, and reduced phagocytosis, as well as efferocytosis at least in pulmonary fibrosis [48]. Matrix stiffness and profibrotic IL-13 influenced alveolar macrophages independently of each other, partly synergistically [48]. For this work, an immortalized murine alveolar macrophage cell line has been used in addition to a hydrogel preparation with different rigidities.

The detailed pathogenesis of biliary atresia (BA) remains unclear so far but represents a kind of inflammatory fibrosis of intra- and extrahepatic bile ducts. In surgically resected tissues from BA patients, IL-13 was visible in 93% of large and micro-bile ducts from human patients—in most of the cases in co-staining with CD45 [49]. Immunohistochemical analyses on this human material suggested an association between IL-13, αSMA, and periostin. Periostin originally detected in mesenchymal cells is secreted into the extracellular matrix. In cultured fibroblasts, periostin expression was enhanced by IL-13 stimulation. Thus, the authors concluded that IL-13 might play a significant role in the fibrotic process of extrahepatic cholestasis, like BA [49].

IL-13 also plays an essential role in the pathophysiology of ulcerative colitis [50]. It is known that IL-13 can disrupt intestinal barrier function by inducing apoptosis and altering the protein composition of tight junctions [34,51]. The activation of the IL-13 receptor a1 by IL-13 increased claudin-2 expression and thus directly or indirectly enhanced intestinal barrier disruption [51]. Since the IL-13 receptor α2 influences other molecules of the tight junctions, the mode of action of IL-13 in inflammatory bowel diseases is very complex [52].

## 3. IL-13 and MASH (Metabolic-Dysfunction-Associated Steatohepatitis)

In the context of *S. japonicum*, a single nucleotide polymorphism (rs1800925T) of the IL13 promoter has already been revealed with increased IL13 gene expression in the liver and an increased risk of pathological liver fibrosis [53]. The authors initially demonstrated that the IL-13 protein was upregulated in fibrotic liver tissue from patients with *S. japonicum* infections. Both in acute infections and in the chronic stage, the functional IL13 promoter polymorphism represented an increased risk for advanced schistosomiasis [53]. The protective properties of this locus with respect to *S. mansoni* and *S. haematobium* could not be confirmed. Different population groups and different stages of infection compared to previous work seem to play a role here [54]. It is still completely unclear why some patients develop severe hepatic complications during schistosomiasis and others living in the same region do not [55].

The impact of IL-13 inhibition on inflammation, fibrosis, metabolism, and tissue repair varies depending on the specific disease context [56]. IL-13 administration activates STAT6, MAPK, and growth factor cascades, leading to fibrosis, allergic inflammation, proliferation, and M2 macrophage polarization based on the specific cell and tissue context [57]. However, IL-13 can also suppress hepatic gluconeogenesis via STAT3 activation [58,59]. In fatty liver disease models, inhibiting IL-13 might worsen insulin resistance, inflammation, and metabolic dysfunction [33]. It disrupts the suppression of gluconeogenesis by blocking STAT3 signaling in hepatocytes. On the other hand, administering IL-13 in these models improves metabolic function by suppressing hepatic gluconeogenesis and lowering glucose production [33]. In adipose tissue, inhibiting IL-13 blocks the polarization of alternatively activated M2 macrophages, worsening inflammation. This occurs through the disruption of STAT6 and PPAR gamma signaling [60]. Conversely, administering IL-13 reduces inflammation by polarizing macrophages to an M2 phenotype via STAT6 activation [60,61]. Compared to wild-type mice, obese mice showed significantly more IL-13 and IL-13 receptors in the normal intestinal mucosa [62]. The addition of IL-13 to colorectal tumor cell lines changed the phenotype of these cells. By means of knockout, the research group was able to work out that IL-13Rα1 was responsible for mucosal proliferation. Thus, a connection between obesity-induced inflammation, the anti-inflammatory IL-13, and colorectal carcinogenesis was established [62].

The question arises whether IL-13 plays a key role in linking inflammation and metabolism. As already stated, this pleiotropic cytokine is involved in the progression from simple steatosis to MASH but also has anti-inflammatory effects. Obesity-associated inflammation in adipose tissue, for example, can be normalized by IL-13. As mentioned, IL-13 is associated with fibrosis, diarrhea, and perianal inflammation in a cholestasis model, and IL13 knockout improves liver function and liver structure, as well as the enteric barrier in mice [34]. From Malaysian hepatitis B cohorts, we know that the serum concentrations of IL-13 were positively correlated with the controlled attenuation parameter (CAP), an ultrasound-based technique for measuring hepatic fat content independently from the presence of fibrosis [63]. In that retrospective analysis, initially a relationship between hepatitis B and liver fibrosis was detected. The liver fat content was measured with the CAP, and the degree of hepatic fibrosis was determined with liver stiffness measurements. The plasma levels of IL-13 were associated with the hepatic fat content independently of other factors. As an incidental finding, hepatitis B patients often suffered from fatty liver [64]. As a conclusion, the authors suggested that IL-13 plays a key role in linking the metabolism and hepatic inflammation [63]. In rats which were fed a special Western diet, the development of MASH was associated with an increase in hepatic IL-13 expression. Exercise treatment, however, reduced this MASH-related IL-13 expression in rats fed a high-fat high-fructose diet [65].

All these results suggest a rather close association between IL-13, MASH, and liver fibrosis. However, the causal relationships remain unclear. Moreover, studies have not yet clarified which hepatic cells react to IL-13 release, induce higher IL-13 expression, and to what extent this occurs. To test whether a cell-specific inhibition of the IL-13 pathway reduces the development of MASLD/MASH, cell-specific knockout mice and mechanistic experiments are necessary. For an illustration, see also Figure 2.

## 4. Different Cells React in Different Ways

As can be seen in Figure 1, IL-13 signaling starts on the cell surface with a multi-subunit receptor through which IL-4 also triggers cellular signaling. This is a heterodimeric receptor complex of the alpha IL-4 receptor (IL-4Rα) and the alpha interleukin-13 receptor (IL-13R1) [66]. The high affinity of IL-13 for IL-13R1 leads to a binding, which increases the likelihood of heterodimer formation with the IL-4R1 [66]. Heterodimerization activates both STAT6 and IRS [67]. STAT6 signaling is important, for example, in the context of the allergic response [20]. Most of the biological effects of IL-13, like those of IL-4, are associated with a single transcription factor, the Signal Transducer and Activator of Transcription 6 (STAT6) [67]. Interleukin-13 and its receptors associated with the α-subunit of the IL-4 receptor (IL-4Rα) facilitate the downstream activation of STAT6 [20,66,67]. In an Egyptian study of 134 male patients with either MASH or MASH-HCC, both high AFP levels and high IL-13 serum levels were measured. The association between IL-13 and a programmed death-ligand 2 polymorphism was predictive of advanced liver fibrosis. High IL-13 levels, however, improved the predictive potential of AFP with regard to carcinogenesis [66]. In patients with MASH, a high expression of IL-13R alpha 2, initially considered as a decoy receptor, seems to be a relevant player in carcinogenesis. IL-13Rα2 was detected in hepatic stellate cells, whereas patients without fatty liver disease did not express this receptor type [68]. The research group also established a MASH model in rats. The treatment of rats with an IL-13 cytotoxin improved dietary-induced MASH fibrosis and associated liver enzymes [68]. Twenty years ago, work was published showing that IL-13 suppresses macrophage production and proinflammatory mediators. The administration of the cytokine IL-13 suppresses the recruitment of neutrophils into the liver and thus hepatic damage, e.g., in the context of ischemia and reperfusion [69]. Furthermore, it has been shown that the IL-13 effect is most likely the result of STAT6 activation [69]. Type 2 innate lymphoid cells regulate epithelial proliferation and tissue repair, whereas inflammatory ILC2s (iILC2s) drive tissue inflammation and injury [70]. The IL-13/IL-4Rα/STAT6 pathway is able to regulate the plasticity of iILC2s, thus affecting and ameliorating epithelial homeostasis and repair, e.g., in experimental biliary atresia [70]. Raabe et al. described a rather new and interesting subset of IL-13-producing ILC3-like cells, which seem to be enriched in human liver and may be involved in the modulation of chronic liver disease [26].

In summary, it seems relatively certain that IL-13 is at least associated with MASH, MASH fibrosis, and MASH-HCC, independently of other variables. Since IL-13 affects hepatic stellate cells, IL-13 inhibition would be associated with a reduction in hepatic fibrosis. A reduction in the matrix build-up inevitably leads to a reduction in the extracellular matrix, i.e., a fibrosis reduction. However, it remains unclear whether IL-13 increases as a consequence of MASH, i.e., as a counter-regulation, or whether it transports the pathogenesis of MASH. Functional studies are therefore needed to investigate the mechanisms of IL-13/STAT6 in MASH, MASH fibrosis progression, and HCC development. Table 1 gives an overview of the variety of IL-13-induced cellular responses with reference to liver cells and fibrosis. Figure 2 gives an overview of the complex interactions of IL-13 in different organs and different cells and possible ways of interaction according to the recent literature.

## 5. Therapeutic Implications

The release of high levels of IL-13 from innate lymphoid type 2 cells promotes cholangiocyte hyperplasia [77]. Beyond cytokine dysfunction in the initial damage of the bile duct epithelium, a disturbed balance of Th1- and Th2-mediated signaling determines the development of chronic liver disease. IL-13 is capable of promoting hepatic fibrogenesis of various etiologies, as shown before, and has been identified as a major pathogenic cytokine in helminth (schistosome)-induced liver disease [53,78]. IL-13 induces alternative activation of macrophages, thereby counteracting Th1-driven inflammatory processes, and is involved, together with galectin-3, in the transition from simple steatosis to MASH [79,80]. Obesity-associated inflammation in adipose tissue and the associated release of inflammatory cytokines can be normalized, for example, by the administration of IL-13. This approach has already been shown to improve the metabolic profile in murine high-fat-diet models (Western diet) [33]. However, high levels of IL-13 are accompanied by negative effects such as fibrosis, diarrhea, and perianal inflammation, the latter being due to a weakening of the intestinal barrier [34,50,73]. IL-13 also plays an important role in metabolic processes in hepatocytes, such as gluconeogenesis [59]. While IL-13 inhibits gluconeogenesis in mouse experiments, a global IL13 knockout produces hyperglycemia, hepatic insulin resistance, and systemic metabolic dysfunction in mice of a C57BL/6 genetic background [59]. These effects were not observed in BALB/c-IL13 knockout mice on a normal diet, as this genetic background is less susceptible to the development of metabolic disease. Hyperglycemia—but not insulin resistance or metabolic dysfunction—was only observed in BALB/c-IL13-knockout mice on a Western diet [59]. Patients with insulin resistance also have elevated serum IL-13 levels, although the level does not correlate with markers of systemic inflammation [81]. The therapeutic application of IL-13 improves post-ischemic gluconeogenesis and hyperglycemia in a rat model [58]. IL-13 is consequently one of the regulators of glucose metabolism that directly inhibits the transcription of hepatic genes encoding enzymes of gluconeogenesis. Anti-inflammatory cytokines such as IL-13 and IL-4 are as important for glucose homeostasis as the proinflammatory cytokines (e.g., TNF-α). According to [59], the pathogenesis of insulin resistance and type 2 diabetes results from a primary defect in the anti-inflammatory arm of the immune system. This defect prevents sufficient hepatic production of IL-13, which is directly required to suppress postprandial hepatic glucose production. However, the overproduction of IL-13 could be detrimental to the liver, as IL-13 promotes the transdifferentiation of hepatic stellate cells to a fibrogenic phenotype [82].

Recently, we analyzed the impact of an IL13 knockout on liver pathology and the intestinal microbiome in Abcb4-knockout mice [34]. In this murine model of cholestasis, we were able to achieve a significant improvement in liver integrity with a global IL13 knockout, at least in this specific animal model [34]. Our results indicate that the observed hepatic effects were due to an improvement in bile duct integrity as well as in the intestinal barrier (reduction in leaky gut syndrome). In addition to the reduction in systemic bile acid concentrations, a reduction in bacterial enterohepatic translocation was shown by our recent work [34]. Building on this basic work on the global IL13 knockout model, we now try to analyze in a more differentiated manner the tissue- and organ-specific contributions of MASH development by cell-type-specific knockouts of the IL-13-receptor with and without the additional systemic application of recombinant IL-13 and IL-13 antibodies.

## 6. Future Directions

Tralokinumab, a biologic available since mid-2021, is an antibody that targets IL-13 specifically. Lebrikizumab (dual IL-4/IL-13 inhibition) is able to block IL-13 as well as IL-4. The mechanism of action of tralokinumab does not include a blockade of IL-4 activity. Whether this could have advantages or disadvantages in terms of therapy in the short or long term is currently unclear. On the efficacy side, clinical studies have clearly shown that this biologic is successfully used in the treatment of the moderate to severe forms of atopic dermatitis [83]. It is also possible to neutralize IL-13 using monoclonal antibodies (mAbs) by blocking IL-13 binding to IL-13 Rα1 and IL-13 Rα2, thus performing a specific inhibition of IL-13.

The successful treatment of MASH in an appropriate mouse model will provide the basis for a translational approach to the treatment of patients with fatty liver disease. Pharmacologically active IL-13 inhibitors, such as pitrakinra (an IL-4 mutant), tralokinumab, or the soluble IL-13Rα2, have already been clinically tested for the treatment of other diseases, such as asthma or in tumor therapy. Further research is urgently needed to address not only different cell types but also liver-derived and liver-migrating IL-13- and IL-13-receptor-expressing cells. Testing whether the cell-specific inhibition of the IL-13 pathway reduces the development of MASLD/MASH in knockout mice is not as easy as expected.

## 7. Conclusions

IL-13 is involved in the processes contributing to the transition from metabolic-dysfunction-associated steatosis to MASH. IL-13 is a Th2-specific cytokine that induces the alternative activation of macrophages, thereby counteracting Th1-driven inflammatory processes. Obesity-associated inflammation in adipose tissue and the associated release of inflammatory cytokines can be normalized, for example, by the administration of IL-13. This approach actually led to an improved metabolic profile in murine high-fat diet models. However, this is accompanied by negative effects, such as fibrosis, diarrhea, and perianal inflammation, the latter side effects being due to a weakening of the intestinal barrier. In the murine model of chronic cholestasis, we were able to achieve a significant improvement in liver function through a global IL13 knockout. Our results show that the observed effects are due to an improvement in the intestinal barrier (reduction in leaky gut syndrome). In addition to molecular biological assays, this was shown by a reduction in bacterial enterohepatic translocation. IL-13 is able to induce distinct cellular functions depending on cell type, organ, and signal transduction pathways.

Furthermore, an IL13 knockout improved/normalized the intestinal microbiome, which is thought to be directly linked to improved barrier function and reduced cholestasis. Thus, it might be speculated that an organ-specific modulation of the IL-13 signaling pathway might be a useful therapeutic approach.

## Figures and Tables

**Figure 1 ijms-24-12884-f001:**
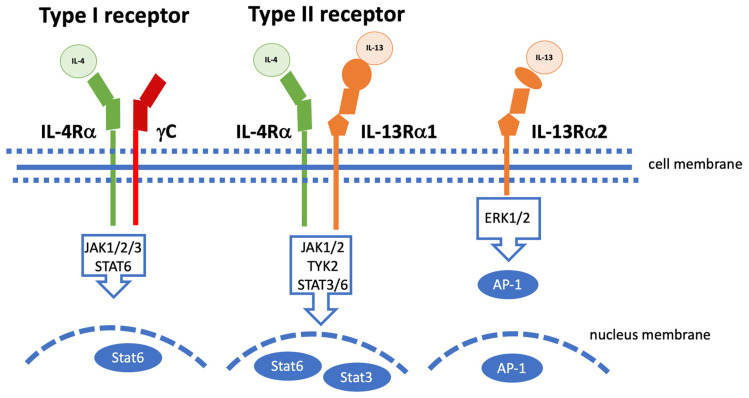
Signaling pathways for IL-13 in various scenarios. Illustration of the three cell membrane receptors that bind IL-4, IL-13, or both according to [20]. IL-4-type I receptor consists of the IL-4Rα subunit and γc. This receptor, which is mainly expressed on hematopoietic cells, binds to IL-4. This binding leads to the activation of JAK1, JAK2, and JAK3 and the subsequent phosphorylation of STAT6. The type II receptor consists of IL-4Rα and IL-13Rα1 (it is found, for example, on smooth muscle cells, fibroblasts, and keratinocytes). Ligand binding of the type II receptor complex leads to activation of JAK1, JAK2, and TYK2 and the subsequent phosphorylation of STAT6 and STAT3. Activated STAT dimers migrate into the nucleus and trigger the activation of downstream genes. IL-13 signals only via the type II receptor. IL-13 also binds to an IL-13Rα2 receptor, whose functions are largely unknown. IL-13 signaling through IL-13Rα2 can lead to STAT6-independent phosphorylation of ERK1/2 and formation of the dimeric transcription factor AP-1. The phosphorylated AP-1 subsequently migrates into the nucleus.

**Figure 2 ijms-24-12884-f002:**
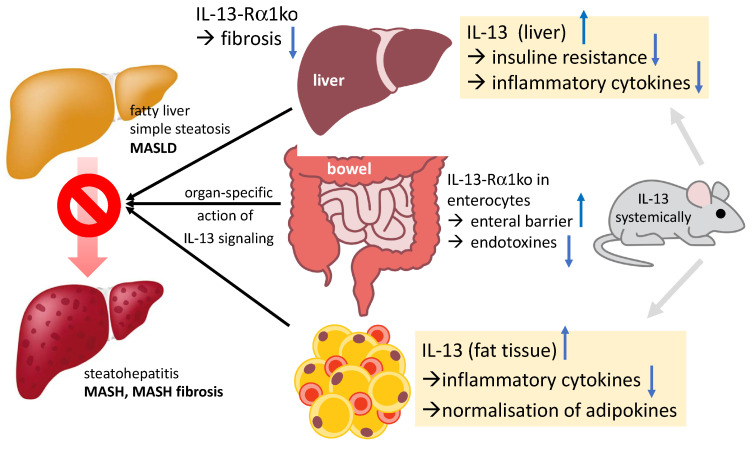
The complex interaction and action of IL-13 in different organs and different cells within one organ is depicted in the figure according to [33,34,59]. Blue arrows depict up- and down-regulation. Ko, knockout; IL, interleukin; HSC, hepatic stellate cells.

**Table 1 ijms-24-12884-t001:** IL-13 is able to induce distinct cellular functions depending on cell type, organ, and signal transduction pathways.

Function of IL-13	Cells	Organ	Literature
Suppressing gluconeogenesis in hepatocytes via STAT3	Group 2 innate lymphoid cells (ILC2s)	Liver pancreas	Fujimoto, *Nat Commun*, 2022 [71]
Induction of hypercholesterolemia No induction of fibrosis	Hepatocytes Hepatocytes	Liver Liver	Low, *Clin Sci*, 2020 [72] Gieseck, *Immunity*, 2016 [73]
Polarization of M2-macrophages inducing white adipose tissue fibrosis	M2-macrophages Macrophages	Oral tissue Adipose tissue	Wang, *J Cell Mol Med*, 2023 [74] Arndt, *Int J Mol Sci*, 2023 [75]
Induction of keloid fibrosis via JAK/STAT6 activation Eosinophil recruitment, liver fibrosis	(Keloid) fibroblasts Liver fibroblasts	Skin Liver	Chao, *JCI Insight*, 2023 [76] Gieseck, *Immunity*, 2016 [73]
Induction of ductular reaction/cholestasis, cholangiocyte differentiation, biliary regeneration	Cholangiocytes Biliary cells	Liver	Gieseck, *Immunity*, 2016 [73]
Intestinal barrier disruption	Mucosal cells	Bowel	Heller, *Gastroenterol*, 2005 [50] Hahn, *Cells*, 2020 [34]

IL-13 is able to target distinct cells, thus driving distinct cellular reactions (left column). IL-13-reacting cells are depicted in the second column, the target organ in the third column, and the references on the right side. The corresponding references are given in square brackets.

## Data Availability

The datasets used during the study are available from the corresponding author on reasonable request.

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
