# Peer review of "Interleukin-13 (IL-13)—A Pleiotropic Cytokine Involved in Wound Healing and Fibrosis"

_ijms, 2023, doi:10.3390/ijms241612884_

Round 1
Reviewer 1 Report
The review titled "IL-13 and its Role in MASH Fibrosis - A Therapeutic Approach? " is an intriguing synthesis work that addresses a significant issue.
The review is well-organized, presenting information in a logical sequence.
It is also informative and offers an excellent scientific synthesis.
Areas for improvement in the review:
The title doesn't accurately convey the purpose of the review.
the author needs to include the signaling pathway for IL13 in various scenarios, including those with and without MASH.
Furthermore, the author should discuss the molecular regulation of IL13 in different metabolic cases, specifically focusing on signaling pathways in obesity.
It is also crucial to consider the molecule regulation between the liver metabolism pathway and inflammation.
I suggest adding the molecular regulation of IL13 with PPARgamma.
Additionally, Table 1 should be in the "Different Cells React in Different Ways" section.
The author must show the major revisions made in the text by highlighting the changes in a different colored text.
It is imperative to consider all these remarks to reinforce the manuscript's quality and conclude more accurately.
Author Response
Reviewer 1
The title doesn't accurately convey the purpose of the review.
Author’s reply: The title has been changed to “Interleukin-13 (IL-13) - a pleiotropic cytokine involved in wound healing and fibrosis.”
The author needs to include the signaling pathway for IL13 in various scenarios, including those with and without MASH.
Author’s reply: Many thanks for the reviewer for this valuable hint. A new figure 1 addressing the receptors´ signaling pathways for IL-13 (Figure 1, signaling pathways for IL-13 in various scenarios) has been included. Thus, illustration of the three cell membrane receptors and signaling pathways have been incorperated into the manuscript. The figure 1 was placed in the introduction of the revised manuscript. Signal transduction depends on the presence of the respective receptors on certain cells. Since the progenitor cells of fibrogenesis in the liver, hepatic stellate cells and fibroblasts, predominantly carry the type II receptor, the latter is essential for MASH and MASH fibrosis. Figure 1 contains the following further information: “IL-4-type I receptor consists of the IL-4Rα subunit and γc. This receptor, which is mainly expressed on hematopoietic cells, binds to IL-4. This binding leads to the activation of JAK1, JAK2, and JAK3 and the subsequent phosphorylation of STAT6. The type II receptor consists of IL-4Rα and the IL-13Rα1 (it is found, for example, on smooth muscle cells, fibroblasts and keratinocytes). Ligand binding of the type II receptor complex leads to activation of JAK1, JAK2, and TYK2 and subsequent phosphorylation of STAT6 and STAT3. Activated STAT dimers migrate into the nucleus and trigger the activation of downstream genes. IL-13 signals only via the type II receptor. IL-13 also binds to an IL-13Rα2 receptor, whose functions are largely unknown.”
Furthermore, the author should discuss the molecular regulation of IL13 in different metabolic cases, specifically focusing on signaling pathways in obesity. It is also crucial to consider the molecule regulation between the liver metabolism pathway and inflammation.
Author’s reply: The molecular regulation of IL-13 in different metabolic diseases especially obesity, liver metabolism pathways, and inflammation has been addressed in the revised manuscript. The following sentences have been added (new manuscript line 227-245, section 3):
“The impact of IL-13 inhibition on inflammation, fibrosis, metabolism, and tissue repair varies depending on the specific disease context [57]. IL-13 administration activates STAT6, MAPK, and growth factor cascades, leading to fibrosis, allergic inflammation, proliferation, and M2 macrophage polarization based on the specific cell and tissue context [58]. However, IL-13 can also suppress hepatic gluconeogenesis via STAT3 activation [59,60]. In fatty liver disease models, inhibiting IL-13 can might worsen insulin resistance, inflammation, and metabolic dysfunction [34]. It disrupts the suppression of gluconeogenesis by blocking STAT3 signaling in hepatocytes. On the other hand, administering IL-13 in these models improves metabolic function by suppressing hepatic gluconeogenesis and lowering glucose production [34]. In adipose tissue, inhibiting IL-13 blocks the polarization of alternatively activated M2 macrophages, worsening inflammation. This occurs through disrupting STAT6 and PPAR gamma signaling [61]. Conversely, administering IL-13 reduces inflammation by polarizing macrophages to an M2 phenotype via STAT6 activation [61,62]. Compared to wild-type mice, obese mice showed significantly more IL-13 and IL-13 receptor in the normal intestinal mucosa [63]. The addition of IL-13 to colorectal tumour cell lines changed the phenotype of these cells. By means of knockout, the research group was able to work out that IL-13R1 was responsible for mucosal proliferation. Thus, a connection between obesity-induced inflammation, the anti-inflammatory IL-13 and colorectal carcinogenesis was established [63].”
I suggest adding the molecular regulation of IL13 with PPARgamma.
Additionally, Table 1 should be in the "Different Cells React in Different Ways" section.
Author’s reply: As you can read above the molecular regulation of IL13 with PPARgamma has also been included into the revised manuscript as well as the corresponding reference. Table 1 was shifted to the section “different cells react in different ways” as suggested.
The author must show the major revisions made in the text by highlighting the changes in a different colored text.
It is imperative to consider all these remarks to reinforce the manuscript's quality and conclude more accurately.
Author’s reply: Many thanks for the reviewer for the kind comments about our manuscript and valuable hints to improve it. All revisions are highlighted by the tool “tracking and marking”. All changes can be tracked using the "MARK up all" tool.
Please see the attachment.

Reviewer 2 Report
Elke Roeb's manuscript summarizes various aspects of IL-13 associated effects on the liver and metabolic liver diseases. The paper is interesting and highlights IL-13's role in MASH fibrosis. However, there are a few issues that need to be addressed.
Major issues include:
1. The author cites several review-type papers as references (Reference #1 #9 #10 #13 #15). It is suggested that the original paper be cited as a reference instead.
2. The author should provide multiple literatures to support their points, not just one literature. For example, from line 48 to line 69 and from line 113 to line 123, the author only cites one literature as a reference.
Minor issues include:
1. The placement of information in line 18-19 needs to be reviewed.
2. The format of the entire paper needs to be consistent. For example, two spaces before a paragraph are needed in line 36, line 48 and line 59.
3. The reference is missing in line 166-169.
4. The reference is missing in line 197-204.
5. It is recommended to relabel Figure A as Figure 1.
Minor editing of English language required
Author Response
Reviewer 2
Major issues include:
- The author cites several review-type papers as references (Reference #1 #9 #10 #13 #15). It is suggested that the original paper be cited as a reference instead.
Author’s reply: References #1 and #9 of the original manuscript (#3 and #11 in the revised manuscript) designate systematic reviews and meta-analyses. They have been cited as themselves to underline the global epidemiology of a specific disease, here NAFLD. As this type of review is the most reliable source of epidemiological data, they are listed here.
The review [10] (#12 in the revised manuscript) discusses the fact, that there is as yet no pharmacotherapy approved for metabolic liver disease. The individual original papers, each dealing with individual substances without marketing authorisation, are beyond the scope of this manuscript. The review in reference [10] is further supported by a national registry study, which also includes or registers the patients' drug treatments [11] (#13 in the revised manuscript).
References [13] and [15] of the original manuscript (that is to say #17 and #29 of the revised manuscript) represent immunological review articles dealing entirely and exclusively with the signal transduction and immunological functions of IL-13. The original reference [15] is a relatively general review article on the role of IL-13 in health and disease and is specified for fibrosis in particular by reference [16] (#30 of the revised form). The author assumes that the citation of the specifically selected review articles (including a very large number of original papers each) seems to make more sense here than the citation of many individual original publications from these reviews. Thus, the numerous original papers are not cited in addition.
Reference numbers have changed in the revised manuscript. To avoid misunderstandings, the reference numbers from the original manuscript are listed here with the new reference numbers in brackets.
- The author should provide multiple literatures to support their points, not just one literature. For example, from line 48 to line 69 and from line 113 to line 123, the author only cites one literature as a reference.
Author’s reply: Many thanks for the important hint. The corresponding statements in the named lines of the original manuscript were supported by further quotations (#16 - #28 of the revised manuscript) and (#38 - #43 of the revised manuscript).
Minor issues include:
- The placement of information in line 18-19 needs to be reviewed.
- The format of the entire paper needs to be consistent. For example, two spaces before a paragraph are needed in line 36, line 48 and line 59.
Author’s reply: Thank you very much for the very thorough review work. Citations for the new nomenclature have been integrated into the revised manuscript. This passage was moved to the introductory section and quoted accordingly ((#1 and #2).
Two spaces before a paragraph have now been integrated as mentioned. The format has been improved accordingly.
- The reference is missing in line 166-169.
Author´s reply: Many thanks for the careful analysis of the manuscript. The missing references (#55, #56) have been integrated into these lines (lines 221-226 of the revised manuscript).
- The reference is missing in line 197-204.
Author’s reply: These issues were discussed in the references [33] and [34] of the initial manuscript. The citations were now integrated in a more specific way including the references #21, #67 and #68 in lines 271-285 of the revised manuscript.
- It is recommended to relabel Figure A as Figure 1.
Author’s reply: Figure A was relabeled respectively and is listed in the revised manuscript as figure 2. A new Figure 1 was additionally included in the manuscript at the suggestion of another reviewer.

Reviewer 3 Report
No comments
Author Response
Reviewer 3
Author`s reply: I thank the reviewer for comment, that the work is a significant contribution to the field.
Reviewer 4 Report
Interleukin-13 (IL-13) is a pleiotropic cytokine involved in wound healing and fibrosis with contradictory effects. While IL-13 may have both protective and detrimental effects on the development of MASH fibrosis, its specific role remains unclear. IL-13 can preserve metabolic functions and promote Th2 polarized inflammation, but it may also contribute to MASH progression by compromising gut barrier function and enhancing hepatic fibrosis. The cellular functions induced by IL-13 depend on various factors such as cell type, organ, and signal transduction pathways. The impact of IL-13 inhibition on inflammation, fibrosis, metabolism, and tissue repair varies depending on the specific disease context. The effects on molecular pathways are likely influenced by the target cell types and organs involved. Further research is needed to elucidate the underlying mechanisms. In contrast, IL-13 administration activates STAT6, MAPK, and growth factor cascades, leading to fibrosis, allergic inflammation, proliferation, and M2 macrophage polarization based on the specific cell and tissue context. However, it can also suppress hepatic gluconeogenesis via STAT3 activation. Inhibiting IL-13 therapeutically reduces fibrosis but may negatively affect metabolic factors and barrier function. While the approach of targeting IL-13 in liver disease is logical, rigorous evaluation of anti-IL-13 therapy in this context requires thorough dose/timing studies, analysis of both inflammation and fibrosis, mechanistic investigations, and consideration of translatability. Further research is needed to fully comprehend the role of IL-13 in MASH fibrosis and its potential therapeutic application.
Moreover:
-Based on the review article, inhibiting or administering IL-13 can have different effects depending on the disease model. In schistosomiasis models of liver fibrosis, inhibiting IL-13 reduces fibrosis by decreasing collagen deposition, alpha-SMA expression, and eosinophil recruitment. This is likely achieved by blocking STAT6 signaling and reducing the levels of TGF-beta and other profibrotic mediators.
-However, in fatty liver disease models, inhibiting IL-13 can worsen insulin resistance, inflammation, and metabolic dysfunction. It disrupts the suppression of gluconeogenesis by blocking STAT3 signaling in hepatocytes. On the other hand, administering IL-13 in these models improves metabolic function by suppressing hepatic gluconeogenesis and lowering glucose production.
-Inflammatory bowel disease models show that inhibiting IL-13 can damage barrier function and increase permeability by disrupting Claudin-2 expression and epithelial tight junctions. The effects of IL-13 administration in inflammatory bowel disease were not mentioned in the review article.
-In asthma models, IL-13 inhibition blocks mucus production, airway remodeling, and hyperresponsiveness, likely through disrupting STAT6 signaling in epithelial cells. However, administering IL-13 induces mucus production and airway hyperresponsiveness by activating STAT6 and increasing the expression of remodeling factors.
-In adipose tissue, inhibiting IL-13 blocks the polarization of alternatively activated M2 macrophages, worsening inflammation. This occurs through disrupting STAT6 and PPAR gamma signaling. Conversely, administering IL-13 reduces inflammation by polarizing macrophages to an M2 phenotype via STAT6 activation.
-Regarding wound healing, inhibiting IL-13 impairs tissue repair, cell proliferation, and barrier function by blocking STAT6, MAPK, and other growth factor cascades. Conversely, IL-13 administration promotes epithelial proliferation and barrier repair through STAT6 activation, enhancing the expression of growth factors.
-In the context of liver fibrosis, the review article did not mention the effects of inhibiting IL-13. However, administering IL-13 promotes stellate cell activation and collagen deposition, likely through STAT6 activation.
…Minor concern: for hepatologists, demand is seen for you to provide a concise table summarizing the key molecular cascades triggered by inhibition and administration of IL-13 in different models: Thank you.
none
Author Response
Reviewer 4
Interleukin-13 (IL-13) is a pleiotropic cytokine involved in wound healing and fibrosis with contradictory effects. While IL-13 may have both protective and detrimental effects on the development of MASH fibrosis, its specific role remains unclear. IL-13 can preserve metabolic functions and promote Th2 polarized inflammation, but it may also contribute to MASH progression by compromising gut barrier function and enhancing hepatic fibrosis. The cellular functions induced by IL-13 depend on various factors such as cell type, organ, and signal transduction pathways. The impact of IL-13 inhibition on inflammation, fibrosis, metabolism, and tissue repair varies depending on the specific disease context. The effects on molecular pathways are likely influenced by the target cell types and organs involved. Further research is needed to elucidate the underlying mechanisms. In contrast, IL-13 administration activates STAT6, MAPK, and growth factor cascades, leading to fibrosis, allergic inflammation, proliferation, and M2 macrophage polarization based on the specific cell and tissue context. However, it can also suppress hepatic gluconeogenesis via STAT3 activation. Inhibiting IL-13 therapeutically reduces fibrosis but may negatively affect metabolic factors and barrier function. While the approach of targeting IL-13 in liver disease is logical, rigorous evaluation of anti-IL-13 therapy in this context requires thorough dose/timing studies, analysis of both inflammation and fibrosis, mechanistic investigations, and consideration of translatability. Further research is needed to fully comprehend the role of IL-13 in MASH fibrosis and its potential therapeutic application.
Moreover:
-Based on the review article, inhibiting or administering IL-13 can have different effects depending on the disease model. In schistosomiasis models of liver fibrosis, inhibiting IL-13 reduces fibrosis by decreasing collagen deposition, alpha-SMA expression, and eosinophil recruitment. This is likely achieved by blocking STAT6 signaling and reducing the levels of TGF-beta and other profibrotic mediators.
-However, in fatty liver disease models, inhibiting IL-13 can worsen insulin resistance, inflammation, and metabolic dysfunction. It disrupts the suppression of gluconeogenesis by blocking STAT3 signaling in hepatocytes. On the other hand, administering IL-13 in these models improves metabolic function by suppressing hepatic gluconeogenesis and lowering glucose production.
Author`s reply: I thank the reviewer for this pointed and focused summary.
-Inflammatory bowel disease models show that inhibiting IL-13 can damage barrier function and increase permeability by disrupting Claudin-2 expression and epithelial tight junctions.
The effects of IL-13 administration in inflammatory bowel disease were not mentioned in the review article.
Author`s reply: I thank the reviewer for this hint. The following sentences and references were included into the revised manuscript (line 203-209):
“IL-13 also plays an essential role in the pathophysiology of ulcerative colitis [51]. It is known that IL-13 can disrupt intestinal barrier function by inducing apoptosis and altering the protein composition of tight junctions [35,52]. The activation of the IL-13 receptor a1 by IL-13 increased claudin-2 expression and thus enhanced intestinal barrier disruption [52]. Since the IL-13 receptor a2 influences other molecules of the tight junctions, the mode of action of IL-13 in inflammatory bowel diseases is very complex [53].”
-In asthma models, IL-13 inhibition blocks mucus production, airway remodeling, and hyperresponsiveness, likely through disrupting STAT6 signaling in epithelial cells. However, administering IL-13 induces mucus production and airway hyperresponsiveness by activating STAT6 and increasing the expression of remodeling factors.
-In adipose tissue, inhibiting IL-13 blocks the polarization of alternatively activated M2 macrophages, worsening inflammation. This occurs through disrupting STAT6 and PPAR gamma signaling. Conversely, administering IL-13 reduces inflammation by polarizing macrophages to an M2 phenotype via STAT6 activation.
-Regarding wound healing, inhibiting IL-13 impairs tissue repair, cell proliferation, and barrier function by blocking STAT6, MAPK, and other growth factor cascades. Conversely, IL-13 administration promotes epithelial proliferation and barrier repair through STAT6 activation, enhancing the expression of growth factors.
-In the context of liver fibrosis, the review article did not mention the effects of inhibiting IL-13. However, administering IL-13 promotes stellate cell activation and collagen deposition, likely through STAT6 activation.
Author`s reply: Thank you for clarifying the IL-13 actions in different tissues. These references were gladly included in the review article and cited accordingly (line 227-245; references #34, #61-#63 of the revised manuscript). Since IL-13 affects hepatic stellate cells, IL-13 inhibition would be associated with a reduction in hepatic fibrosis. Among other things, this is also related to the permanent matrix build-up and matrix degradation in the liver. A reduction in matrix build-up inevitably leads to a reduction in the extracellular matrix, i.e. a fibrosis reduction. This is also shown in figure 2 (figure A of the initial manuscript), among others. The following sentences were integrated into the revised manuscript, line 317-320: “Since IL-13 affects hepatic stellate cells, IL-13 inhibition would be associated with a reduction in hepatic fibrosis. Since a reduction in matrix build-up inevitably leads to a reduction in the extracellular matrix, i.e. a fibrosis reduction.”
…Minor concern: for hepatologists, demand is seen for you to provide a concise table summarizing the key molecular cascades triggered by inhibition and administration of IL-13 in different models: Thank you.
Author`s reply: Thank you for the valuable advice. As table 1 together with the new figure 1 (recommended by another reviewer) already contain much of the content of the proposed additional table, table 1 has been expanded to include the missing findings on IL-13.
The inserted references allow the interested reader to read further

Reviewer 5 Report
The reviewed work is written very correctly. There are no major errors in it, figures and text correspond well to the current state of knowledge. I have no comments on this publication.
Round 2
Reviewer 1 Report
In this version of the review, “Interleukin-13 (IL-13) - a pleiotropic cytokine involved in wound healing and fibrosis.” We can see an acceptable evolution compared to the first version because it has become more structured with more explanation.
the authors have relatively taken the reviewer's remarks and suggestions into consideration, which has positively impacted the quality and consistency of the article.
with this version, the article shows an excellent scientific level and represents an added value in the interested research topics
the article is accepted for me with this version